# A Global Neighborhood with Hill-Climbing Algorithm for Fuzzy Flexible Job Shop Scheduling Problem

**Juan Carlos Seck-Tuoh-Mora** *,†, **Nayeli Jazmín Escamilla-Serna** †, **Leonardo Javier Montiel-Arrieta**, **Irving Barragan-Vite** and **Joselito Medina-Marin**

Área Académica de Ingeniería, Instituto de Ciencias Básicas e Ingeniería, Universidad Autónoma del Estado de Hidalgo, Carr. Pachuca-Tulancingo km. 4.5, Pachuca 42184, Hidalgo, Mexico
* Correspondence: jseck@uaeh.edu.mx
† These authors contributed equally to this work.

**Abstract:** The Flexible Job Shop Scheduling Problem (FJSSP) continues to be studied extensively to test new metaheuristics and because of its closeness to current production systems. A variant of the FJSSP uses fuzzy processing times instead of fixed times. This paper proposes a new algorithm for FJSSP with fuzzy processing times called the global neighborhood with hill-climbing algorithm (GN-HC). This algorithm performs solution exploration using simple operators concurrently for global search neighborhood handling. For local search, random restart hill-climbing is applied at each solution to find the best machine for each operation. For the selection of operations in hill climbing, a record of the operations defining the fuzzy makespan is employed to use them as a critical path. Finally, an estimation of the crisp makespan with the longest processing times in hill climbing is made to improve the speed of the GN-HC. The GN-HC is compared with other recently proposed methods recognized for their excellent performance, using 6 FJSSP instances with fuzzy times. The obtained results show satisfactory competitiveness for GN-HC compared to state-of-the-art algorithms. The GN-HC implementation was performed in Matlab and can be found on GitHub (check Data Availability Statement at the end of the paper).

**Keywords:** job shop scheduling; fuzzy processing times; global search; hill climbing; critical path

**MSC:** 68T20; 68W50; 90C59

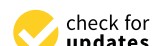



## 1. Introduction

The Flexible Job Shop Scheduling Problem (FJSSP) is a problem that continues to be widely studied to test new combinatorial metaheuristics because of its level of complexity as well as its closeness to today's production systems, where several machines can process the same operations at different processing times [1]. A variant of the FJSSP is to use fuzzy processing times instead of fixed times, which makes it closer to reality. This problem is known as the Fuzzy Flexible Job Shop Scheduling Problem (FFJSSP) [2].

Currently, new algorithms for the FFJSSP continue to be proposed based on population metaheuristics, where the interaction between the members of a population serves to exchange information and refine solutions [3]. In most of these proposals, a local search method is also used to improve these solutions, where the value to be minimized is the makespan or fuzzy processing time in which all operations are completed [4].

This paper proposes a new method to solve the FFJSSP called the global neighborhood with hill-climbing (GN-HC) algorithm. This algorithm performs solution exploration using a global search neighborhood supported by insertion, exchange and path-relinking operations applied concurrently. For local search, hill climbing with random restart is applied on each solution where critical operations are first detected and one of them is randomly chosen and switched to a different processing machine. The critical operations

were selected backtracking from the last operation defining the makespan, going through all preceding operations until the initial operation is reached. In order to speed up the processing time of hill climbing, the new crisp makespan is estimated using the longest of the fuzzy processing times to find the best machine for each operation.

The contribution of this work lies in defining an algorithm whose global search for solution exploration is based on the concurrent application of elementary and well-known operations. The exploitation by hill climbing also consists of simple operators that define a robust and easy-to-implement method, improved by the selection of critical operations and the estimation of makespan to speed up the computational execution.

The GN-HC shows competitive and satisfactory results compared with eight other recently published methods taking the same benchmark set with six cases [5] increasing in the number of operations and machines.

The paper is organized as follows. Section 2 presents a brief literature review of current trends in the study of the FFJSSP. Section 3 formally defines the FFJSSP, the constraints used in the model and how to compare fuzzy times for minimization actions. Section 4 describes the different exploration and exploitation operations defining the GN-HC algorithm, as well as their computational complexity. Section 5 explains how the GN-HC parameters were tuned and the experiments performed on the five test problems to compare the proposed algorithm against the other six methods. Finally, Section 6 gives the conclusions and prospects for possible future work regarding this research.

## 2. Literature Review

The job shop scheduling problem (JSSP) is one of several types of [6] scheduling problems. The flexible job shop scheduling problem (FJSSP) is a generalization of the JSSP where each operation of each one of the jobs must be assigned to a machine from a set of feasible machines and it is desired to obtain a satisfactory schedule where all the operations of all the jobs are executed in the briefest possible time.

The FJSSP was first addressed in [7] under a two-job scheme and a polynomial algorithm to solve it. The FJSSP has gained the attention of both researchers and industries interested in solving this type of problem since it is ideal for real applications and the amount of study directed at it has been increasing in the last decade [8]. However, due to the difficulty of finding a solution satisfying all the conditions of the FJSSP, a wide variety of methods have been proposed, among them evolutionary and swarm intelligence algorithms [9].

One of the assumptions in the FJSSP is to consider the deterministic parameters. Nevertheless, in reality, this does not happen due to different circumstances such as machine failures, power interruptions and varying delivery times, among others, creating an environment of uncertainty and inaccuracy in the values of the parameters that are not manageable by the proposed methods to solve the deterministic version of the FJSSP. To deal with uncertain and imprecise processing times, fuzzy scheduling has been an approach that has gained interest in estimating the completion time of jobs [10].

The fuzzy programming approach consists of considering the parameters of the problem as fuzzy numbers [11]. In general, the processing times of the operations are the parameter to be considered fuzzy functions in the FJSSP, giving rise to the fuzzy flexible job shop scheduling problem (FFJSSP). Genetic algorithms are one of the classic metaheuristics used to solve the FFJSSP.

In [12], a co-evolutionary genetic algorithm minimizes the fuzzy makespan by using a new crossover operator and a modified tournament selection. At [13], the authors employed a genetic algorithm-based approach, proposing a new chromosome structure to avoid losing or destroying elite solutions along with the principles of immunity and entropy to maintain the diversity of individuals and overcome premature convergence. The hybridization of a genetic algorithm with tabu search and a heuristic seeding to minimize the fuzzy makespan was used in [4]. In [14], an adaptive genetic algorithm is proposed using a cloud computing method for crossover and mutation operators.

Another of the methods used repeatedly to address the FFJSSP is differential evolution, as in [15], where they use a modified differential evolution algorithm to establish a schedule that allows more work to be completed before the due date. In [16], a hyper-heuristic algorithm is used in which a set of low-level heuristics is created and differential evolution is used as a high-level strategy to handle these heuristics. In [17], a differential evolution algorithm is proposed based on the determination of the parameters of the fuzzy membership functions according to the calculation of the maximum satisfaction rate to measure the effectiveness of the programming. In [18], the authors propose a new selection operator to improve the classical differential evolution algorithm.

Swarm intelligence-based algorithms have also been widely used to solve the FFJSSP as in [19–21], for instance, bee colony-based algorithms [22–24] and cooperative algorithms, [12,25,26]. A set of five population-based metaheuristics are compared in [27], where the PSO algorithm is suitable for solving both the FJSSP and the FFJSSP. Other proposals include a distribution estimation algorithm [28] to model the probability distribution of the solution space, biogeography-based hybrid optimization [29], hybrid multiverse optimization [30], a discrete flower pollination algorithm [31] and a hyper-heuristic based on backward search [32].

All the previous works show a strong tendency to propose hybrid algorithms combining different methods for the actions of exploring the solution space and exploiting the solutions reached to optimize FFJSSP instances. This trend is followed by the algorithm proposed in this work, presenting a hybrid method that performs the exploration using classical operators concurrently to generate a neighborhood that chooses the best neighbor as the new solution and the exploitation with a random-restart hill-climbing using critical operations and the estimation of the makespan to yield a fast exploitation of solutions.

## 3. Problem Formulation

The flexible job shop scheduling problem (FJSSP) consists of a set of $n$ jobs $J = J_1, J_2, \ldots J_n$ and a set of $m$ machines $M = M_1, M_2, \ldots M_m$. Each job $J_i$ is composed of a sequence of operations $O_{J_i} = \{O_{i,1}, O_{i,2}, \ldots, O_{i,n_i}\}$ where $n_i$ is the number of operations of job $J_i$. Each operation $O_{i,j}$ can be processed by one machine from a set of feasible machines $M_{i,j} \subseteq M$, for $1 \leq i \leq n$ and $1 \leq j \leq n_i$ [32]. In the classical definition of the FJSSP, the processing time of $O_{i,j}$ on the $M_k$ machine is a fixed value. However, this assumption may be insufficient for many applications since there may be a different processing time for the same type of product due to the nature of the process, the technology implemented, or the human factor.

To make the definition of the FJSSP more realistic for cases where a fixed processing time cannot be assured, one option is to use a fuzzy processing time, where the processing time has a minimum, most probable and maximum value. This variant will be defined as a fuzzy FJSSP (FFJSSP). Thus, the processing times are handled using fuzzy values and fuzzy operations [32].

The processing time of the $O_{i,j}$ operation on the $M_k$ machine is represented as a fuzzy number $TF = (t^1_{i,j,k}, t^2_{i,j,k}, t^3_{i,j,k})$. The fuzzy completion time of $O_{i,j}$ is represented as a fuzzy number $C_{i,j} = (C^1_{i,j}, C^2_{i,j}, C^3_{i,j})$ with $C^1_{i,j}$ as the shortest possible completion time, $C^2_{i,j}$ is the most probable completion time and $C^3_{i,j}$ is the longest possible completion time. For this situation, a solution to an FFJSSP instance determines the assignment of machines to each operation and defines the appropriate sequence of operations to complete all jobs that minimize the maximum completeness time defined in Equation (1).

$$C_{max} = \max\{C_i \text{ for } 1 \leq i \leq n\} \tag{1}$$

where $C_i$ is the fuzzy completeness time of the complete job $J_i$.

In order to correctly calculate the fuzzy times for the computation of completeness times, one needs to define operations such as addition, ranking and a maximum of two fuzzy numbers. Addition will calculate the completeness time of an operation, ranking will compare two fuzzy numbers in order to select the starting time of an operation and order

fuzzy numbers for an elitist selection of solutions. For two fuzzy numbers $X = (x_1, x_2, x_3)$ and $Y = (y_1, y_2, y_3)$, addition is defined as $X + Y = (x_1 + y_1, x_2 + y_2; x_3 + y_3)$. For ranking two fuzzy numbers, the following criteria are used [17].

$$
\begin{aligned}
Z_1(X) &= (x_1 + 2x_2 + x_3)/4 \\
Z_2(X) &= x_2 \\
Z_3(X) &= (x_3 - x_1)
\end{aligned} \tag{2}
$$

The maximum between $X, Y$ will be the one with the highest ranking, i.e., $X > Y$ implies that $Z_1(X) > Z_1(Y)$, or if $Z_1(X) = Z_1(Y)$ then $Z_2(X) > Z_2(Y)$, or in the latter case if $Z_2(X) = Z_2(Y)$ then $Z_3(X) > Z_3(Y)$ [22,27,28].

In this work, the objective function to minimize is the fuzzy makespan defined in Equation (1). In the operation of an FFJSSP instance, the following conditions are considered: (1) an operation cannot be interrupted while being processed by a machine; (2) a machine can process at most one operation; (3) once the order of operations is determined, it cannot be changed; (4) no decompositions in the machines are considered; (5) the jobs are independent of each other; (6) the machines are independent of each other; (7) the time used for machine setup and transfer of operations between machines is negligible.

## 4. Global Neighborhood with Hill-Climbing Algorithm

The global neighborhood with hill-climbing algorithm (GN-HC) is a population metaheuristic divided into two stages. The first one is to generate a global-search neighborhood for each solution or smart cell in the population by exchanging information with other smart cells to generate a set of new solutions; the best one will be chosen to update the position of the original smart cell. The second stage consists of performing a local search that takes only each smart cell's information to improve its position. This local search is based on applying random-restart hill-climbing.

### 4.1. Encoding and Decoding Smart Cells

A smart cell consists of two strings $O_s$ and $M_s$, where $O_s$ defines the order in which the machines process operations. $O_s$ is represented as a permutation with repetitions of the $J_n$ jobs, where each job is repeated as many times as it has operations. $M_s$ defines the assignment of a machine for each operation in $O_s$. The first $n_1$ positions of $O_s$ specify the machines of the operations for $J_1$, the following $n_2$ positions define the machines assigned to the operations in $J_2$ and so on. The goal of the problem is to find the best smart cell that minimizes the fuzzy makespan $C_{max}$.

Figure 1 shows an example of an FFJSSP with two jobs, two operations per job and two machines. To decode the solution, the sequence $O_s$ indicates that operation 1 of job 2 is scheduled first, then operation 2 of job 2 and so forth. The sequence $M_s$ indicates that machine 1 is the one that processes operation 1 of job 1, then the same machine will process operation 2 of job 1, etc.

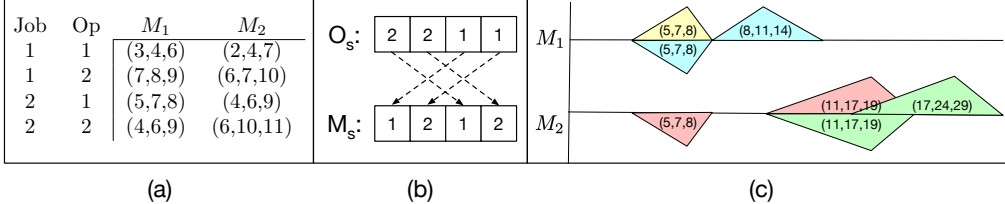

**Figure 1.** FFJSSP example. (**a**) Description of the problem and the fuzzy processing times. (**b**) Decoding of the $O_s$ and $M_s$ sequences. (**c**) Fuzzy Gantt chart corresponding to the solution specified by $O_s$ and $M_s$.

The GN-HC starts by generating a random population $S$ with $S_n$ smart cells. Each solution $s \in S$ consists of two strings $O_s$ and $M_s$. From the beginning and at each iteration

of the algorithm, the best smart cells are selected using elitism and tournament. For elitism, the best $b$ smart cells from the current population are selected to be part of the population in the next iteration. These $b$ best solutions have the lowest fuzzy makespan following the criteria of Equation (2). A tournament determines the remaining $S_n - b$ solutions by selecting $S_n - b$ random pairs of smart cells in $S$ and taking from each pair the one with the lowest fuzzy makespan.

### 4.2. Global Neighborhood

The GN-HC first generates a global neighborhood for each smart cell, using insertion, swapping and path-relinking operations on the sequence of operations $O_s$. On the sequence of assigned machines $M_s$, a mutation is performed selecting another viable machine.

The insertion consists of changing the position of an element of $O_s$ by moving the operations between the original position and the new position of the selected operation. Swapping consists of randomly selecting two elements of $O_s$ and exchanging their positions.

For path-relinking, two smart cells are selected and their operation strings $O_{s_1}$ and $O_{s_2}$ are taken. Then, a sequence of strings leading from $O_{s_1}$ to $O_{s_2}$ is made by swapping the elements of $O_{s_1}$ taken from right to left that have a different value in $O_{s_2}$ for the same position. Each swapping defines a new string; thus, a sequence of strings from $O_{s_1}$ to $O_{s_2}$ is constructed. One of these strings is randomly taken as the new sequence $O_{s_1}$.

For machine assignment strings $M_s$, a random mutation is performed by choosing a random position and changing the assigned machine to a feasible one. For the $O_s$ and $M_s$ sequences used in Figure 1, the actions of these operators are illustrated in Figure 2.

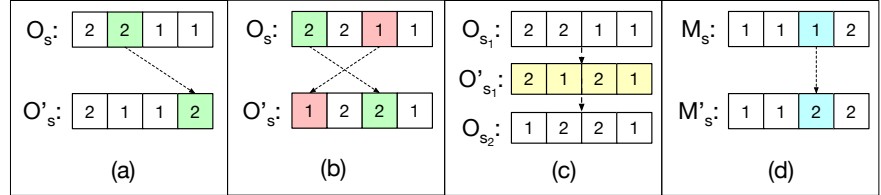

**Figure 2.** Operators used in the global neighborhood. (**a**) Insertion. (**b**) Swapping. (**c**) Path-relinking. (**d**) Mutation of $M_s$.

For solution exploration, the GN-HC makes a neighborhood for each smart cell where randomly $l$ neighbors are generated, each one obtained by insertion with probability $\alpha_I$, by swapping with probability $\alpha_S$ or by path-relinking with probability $\alpha_P$ in order to obtain a new sequence $O_s$, such that $\alpha_I + \alpha_S + \alpha_P = 1$. For each new neighbor, a mutation is applied on $M_s$ with probability $\alpha_M$. From all these $l$ neighbors, the one with the lowest fuzzy makespan $C_{max}$ is chosen, which will update the original smart cell. The neighborhood used for the global search is illustrated in Figure 3. Neighbors are generated by random selection of insertion, swapping and path-relinking to modify $O_s$ and by mutation of the $M_s$ sequence. The best neighbor is selected as the new smart cell.

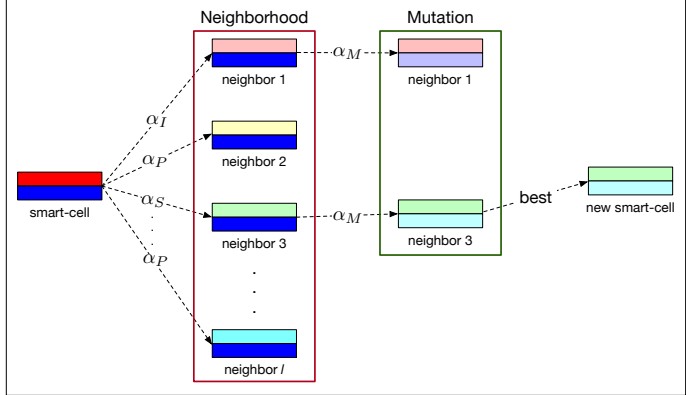

**Figure 3.** Global neighborhood for exploration of the search space.

### 4.3. Local Search

A local search based on hill climbing is performed to improve each smart cell. The search consists of finding the critical path of each smart cell, choosing a critical operation randomly and changing its machine at random.

The strategy to improve the local search performance involves applying hill climbing for $H_n$ iterations over a smart cell. If the smart cell has not improved after $R_n < H_n$ iterations, then one of the previous $R_n$ solutions is randomly taken as the new restart solution to continue the local search for the rest of the iterations. This strategy has been successfully tested in other instances of the FJSSP [33], being effective and straightforward to implement.

Since we are working with fuzzy times, it is impossible to talk about calculating a critical path in the classical sense [4]. The approach taken in this work is to save the antecedent operation selected to define the completion time of each operation. If an operation has no antecedent one, that record is stored as 0. This record is made when the fuzzy makespan is computed and does not require additional computational time. To find the critical operations, only backward tracing is conducted by selecting the operation that defines the fuzzy makespan and going back in the chain of previous operations until reaching an operation without antecedent. These operations are the ones that will be selected for hill climbing since if they do not move, the current makespan will not change.

Figure 4 shows the operations that define the fuzzy makespan for the example in Figure 1 and how the makespan changes when a critical operation is assigned to another feasible machine.

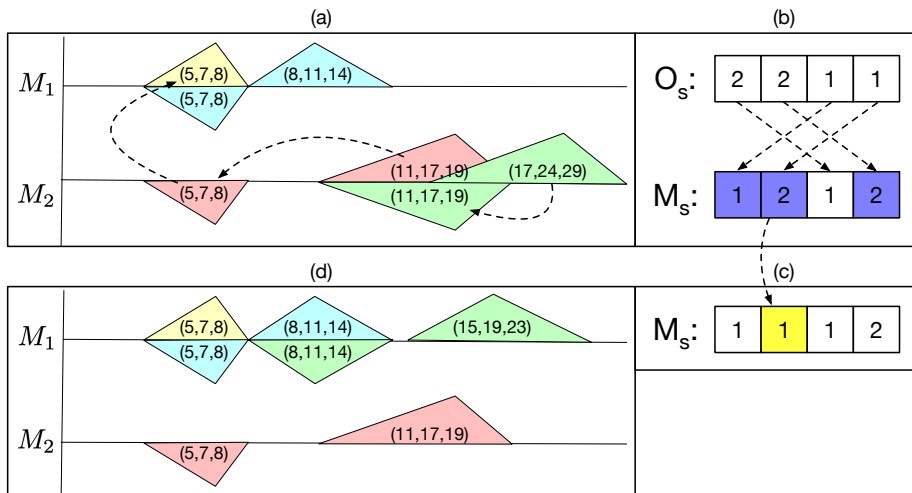

**Figure 4.** Machine change in a critical operation during the hill climbing. (**a**) Backtracking of critical operations. (**b**) Machines assigned to critical operations. (**c**) Machine change in a critical operation. (**d**) New fuzzy makespan.

In order to speed up the hill-climbing process, makespan estimation is used as explained in [34] using the maximum values of the fuzzy processing times to obtain a crisp estimate of each maximum makespan obtained when switching a machine of a critical operation. For this case, the crisp makespan estimation depends on the record of the end and tail times of each critical operation. This record is made on the maximum processing times as the fuzzy makespan is calculated; therefore, it does not imply an increase in the algorithm's computational complexity.

If the estimated crisp makespan is greater than the maximum makespan of the smart cell, the new solution is discarded. In this way, only the complete fuzzy makespan is calculated for the new solutions whose estimated crisp makespan improves the smart cell's maximum makespan, reducing the local search's execution time.

### 4.4. Complete GN-HC Algorithm

The complete GN-HC pseudocode is presented in Algorithm 1. The algorithm parameters are the number of smart cells $S_n$, the number of neighbors per smart cell $l$ and the number of iterations $G_n$. First, the random population of smart cells is generated, the fuzzy makespan of each smart cell is calculated and the optimization loop starts. In the loop, a population refined by elitism and tournament is chosen. For each smart cell in the new population, a global neighborhood of new solutions is calculated by randomly selecting $l$ operations between insertion, swapping and path-relinking. The best solution with the minimum makespan between the original smart cell and its neighborhood is selected as the new smart cell.

Once the global search is finished, hill climbing is performed on the smart cell. The process is terminated by fulfilling a number of total iterations or by a number of iterations with no change in the best solution.

---

**Algorithm 1:** General description of the GN-HC

---

**Result:** Best smart cell

Establish the parameters of the GN-HC;

Initializing the population of smart cells with $S_n$ randomly generated solutions;

Evaluate each smart cell to obtain its fuzzy makespan;

**do**

> Generate a new population selecting the best smart cells by elitism and tournament;
>
> For each non-elitist smart cell, generate a neighborhood (using the insertion, swap and path relinking operators) and take the best neighbor as the new smart cell;
>
> For each smart cell, a random restart hill-climbing is performed to improve its fuzzy makespan;

**while** *(Iterations less than $G_n$ or stagnation iterations less than $S_b$.)*;

Return the smart cell with minimum fuzzy makespan;

---

### 4.5. Computational Complexity of the GN-HC

The execution time will not be taken to compare the computational complexity of the GN-HC against other recent methods since it varies depending on the language, the computer characteristics and the programming skills to implement each algorithm. In our case, the computational complexity of the proposed algorithm will be analyzed.

The GN-HC is divided into two parts: the global search based on a neighborhood generated by insertion, swapping and path-relinking; a machine mutation; and the local search by a random restart hill climbing. These operations are executed at each iteration, which first performs selection by elitism and tournament, which involves ordering the population by its fuzzy makespan. This process has complexity $\mathcal{O}(S_n \times \log S_n)$.

Let $o = \sum_{i=1}^{n} |O_{J_i}|$ be the number of total operations of an FFJSSP instance, then each operation of the global search (insertion, swapping, path-relinking and machine mutation) has complexity $\mathcal{O}(o)$. The global search is applied to each smart cell in order to form a neighborhood of $l$ new solutions. Thus, at each iteration of the GN-HC the complexity is $\mathcal{O}(l \times S_n \times o)$.

For the local search, $H_n$ iterations are performed by modifying the machines of the critical operations of the smart cell. These critical operations are computed by backtracking the record of the operations defining the fuzzy makespan. Consequently, forming the critical path has complexity at most $\mathcal{O}(o)$. Hence, at each iteration of GN-HC, the local search applies hill-climbing to each smart cell for $H_n$ iterations, obtaining a total complexity of $\mathcal{O}(S_n \times H_n \times o)$.

Thus, for each GN-HC iteration, the total complexity is given by $\mathcal{O}(S_n \times \log S_n) + \mathcal{O}(l \times S_n \times o) + \mathcal{O}(S_n \times H_n \times o)$ equivalent to $\mathcal{O}(S_n(\log S_n + ((l + H_n) \times o)))$. This complexity is similar to those exhibited by recent algorithms, e.g., the backtracking search-based

hyper-heuristic (BS-HH) [32] and the hybrid multi-verse optimization (HMVO) [30], which shows that the proposed algorithm is competitive with recent methods in terms of computational complexity.

## 5. Experimental Results

In this experiment, six instances of the FFJSSP proposed in [5,19] are taken to test the effectiveness of GN-HC. These instances have size ranging from 10 jobs, 10 machines and 40 operations to 15 jobs, 10 machines and 80 operations. For each instance, GN-HC was run 30 times independently, using at most $G_n = 500$ optimization iterations, comparable to experiments performed on the other algorithms.

*GN-HC Parameters*

A preliminary analysis was performed to define the GN-HC parameters using Case 1 of [5] to select the best values. The works presented in [32,35] were used to specify the number of generations and the population size, because they are recent works that show significant effectiveness in minimizing FFJSSP instances.

In [32], $G_n$ and $S_n$ are taken as 400 and 200, respectively; meanwhile, in [35], $G_n$ goes from 500 to 800 and $S_n$ goes from 60 to 80. For the GN-HC parameters, intermediate values with these works were tested to show that the proposed algorithm can obtain competitive results with a simple implementation. Thus, 400 and 500 were tested for $G_n$ and 80 and 100 for $S_n$.

The number of neighbors $l$ for the global neighborhood was 3 and 5. To form the neighborhood, probability combinations $(\alpha_I, \alpha_S, \alpha_P)$ with $(0.5, 0.25, 0.25)$, $(0.25, 0.5, 0.25)$ and $(0.25, 0.25, 0.5)$ were tested. The mutation probability $\alpha_M$ was tested with values 0.1 and 0.2. To control the stagnation parameter $S_b$, values 100 and 200 were sampled. For the elitist proportion of solutions in the smart cell population, $E_p$ values of 0.025 and 0.05 were considered. For the random-restart hill climbing, values 100 and 150 for $H_n$ and 10 and 15 for $H_r$ were tested.

In total, there were 768 combinations of parameters. For each one, 10 independent runs were performed and the combination with the best average results was selected. With this experimentation, the parameters of the proposed algorithm are as follows:

- Number of optimization generations $G_n = 500$.
- Number of smart cells $S_n = 80$.
- Neighborhood size $l = 5$.
- Probabilities $(\alpha_I = 0.5, \alpha_S = 0.25, \alpha_P = 0.25)$.
- Probability $\alpha_M = 0.1$.
- Number of stagnation generations $S_b = 100$.
- Proportion of elitist solutions $E_p = 0.05$.
- Hill-climbing iterations $H_n = 150$.
- Iterations to restart the hill climbing $H_r = 15$.

*Comparative Results*

The algorithm is compared with eight other algorithms recognized for their excellent results for this type of problem. The algorithms used for the comparison of results are the backtracking search-based hyper-heuristic (BS-HH) [32], the co-evolutionary genetic algorithm (CGA) [12], the hybrid genetic tabu search (HGTS) [4], the hybrid multi-verse optimization (HMVO) [30], the hybrid QPSO (HQPSO) [35], the improved artificial bee colony algorithm (IABC) [23], swarm-based neighbourhood search algorithm (SNSA) [19] and the effective teaching–learning-based optimization algorithm (TLBO) [36]. The results are shown in Table 1 and the results for the other algorithms are taken directly from the literature.

The GN-HC was programmed in Matlab R2015a(TM) using a 2.3 GHz Intel Xeon W computer and 128 GB of RAM. Table 1 shows the best, worst and average fuzzy makespan achieved by the GN-HC in 30 independent runs for each of the five cases. In each case,

the number of jobs and the number of machines ($n \times m$) is specified. For each algorithm, the ranking obtained compared to the other published results is shown. The algorithm with the best average fuzzy makespan reported so far is shown in bold and the ranking obtained by GN-HC in each case is shown in red. Some methods do not report results for Case 6; in this event, the corresponding cell in Table 1 is left empty. The HGTS only reports average fuzzy makespans; these values are used to rank HGST in the Average column.

**Table 1.** Experimental results of GN-HC and its comparison with various metaheuristics for Lei instances.

| Problem | Algorithm | Average | Rank | Best | Rank | Worst | Rank |
|---------|-----------|---------|------|------|------|-------|------|
| Case 1 | **BS-HH** | (18.5,26.9,36.0) | 1 | (18,26,36) | 1 | (18,27,37) | 1 |
| (10 × 10) | CGA | (23.1,33.1,43.4) | 7 | (21,29,41) | 3 | (25,37,47) | 7 |
| | GN-HC | (20.2,28.2,38.5) | 3 | (21,28,37) | 2 | (22,30,39) | 3 |
| | HGTS | (——,28.5,——) | 5 | | | | |
| | HMVO | (21.9,28.0,38.1) | 4 | (21,28,37) | 2 | (19,28,39) | 2 |
| | HQPSO | (21.0,28.0,37.0) | 2 | (21,28,37) | 2 | (21,28,37) | 2 |
| | IABC | (20.1,29.4,40.3) | 6 | (19,28,39) | 2 | (22,30,42) | 4 |
| | SNSA | (21.9,31.8,41.2) | 8 | (21,29,42) | 4 | (23,33,42) | 6 |
| | TLBO | (20.3,29.9,40.9) | 7 | (19,28,39) | 2 | (21,32,42) | 5 |
| Case 2 | **BS-HH** | (28.8,40.0,52.5) | 1 | (29,39,51) | 1 | (32,39,54) | 1 |
| (10 × 10) | CGA | (35.0,47.1,60.6) | 8 | (32,47,57) | 4 | (38,49,64) | 6 |
| | GN-HC | (32.0,46.0,57.3) | 4 | (30,45,58) | 2 | (35,46,57) | 3 |
| | HGTS | (——,45.2,——) | 3 | | | | |
| | HMVO | (30.0,45.0,58.0) | 2 | (30,45,58) | 2 | (30,45,58) | 2 |
| | HQPSO | (30.0,45.0,58.0) | 2 | (30,45,58) | 2 | (30,45,58) | 2 |
| | IABC | (32.3,46.2,57.3) | 5 | (33,45,58) | 5 | (35,46,57) | 3 |
| | SNSA | (34.9,46.4,60.5) | 7 | (35,43,60) | 3 | (38,48,63) | 5 |
| | TLBO | (32.6,46.4,58.5) | 6 | (30,45,58) | 2 | (36,49,63) | 4 |
| Case 3 | **BS-HH** | (29.5,42.5,55.9) | 1 | (30,42,54) | 1 | (28,44,56) | 1 |
| (10 × 10) | CGA | (36.4,50.8,66.0) | 7 | (34,47,63) | 8 | (38,53,71) | 6 |
| | GN-HC | (30.8,44.1,59.2) | 4 | (29,44,59) | 4 | (32,44,59) | 3 |
| | HGTS | (——,43.5,——) | 2 | | | | |
| | HMVO | (31.0,43.8,58) | 3 | (29,44,58) | 3 | (32,44,59) | 3 |
| | HQPSO | (29.2,43.5,58.2) | 2 | (28,43,59) | 2 | (29,44,58) | 2 |
| | IABC | (31.8,45.8,59.6) | 5 | (31,45,57) | 5 | (33,47,63) | 4 |
| | SNSA | (35.6,51.1,67.2) | 8 | (36,46,62) | 7 | (36,54,75) | 7 |
| | TLBO | (31.5,46.7,62.2) | 6 | (30,45,60) | 6 | (33,50,70) | 5 |
| Case 4 | **BS-HH** | (21.5,33.0,46.3) | 1 | (21,32,47) | 1 | (24,33,46) | 1 |
| (10 × 10) | CGA | (27.4,40.4,55.0) | 8 | (26,37,51) | 7 | (29,42,59) | 6 |
| | GN-HC | (24.5,34.7,47.6) | 5 | (24,34,47) | 4 | (27,37,49) | 3 |
| | HGTS | (——,34.2,——) | 4 | | | | |
| | HMVO | (22.5,34.0,48.0) | 3 | (25,33,47) | 3 | (25,34,48) | 2 |
| | HQPSO | (22.6,33.6,47.5) | 2 | (23,33,47) | 2 | (23,34,48) | 2 |
| | IABC | (24.1,36.1,50.9) | 6 | (25,34,49) | 5 | (24,38,55) | 4 |
| | SNSA | (27.9,40.9,56.1) | 9 | (26,39,53) | 8 | (31,43,56) | 7 |
| | TLBO | (24.9,36.5,50.8) | 7 | (21,36,50) | 6 | (26,40,57) | 5 |
| Case 5 | BS-HH | (35.3,52.6,73.0) | 3 | (36,52,69) | 2 | (33,53,77) | 1 |
| (15 × 10) | CGA | (47.0,65.4,86.0) | 8 | (42,62,82) | 7 | (49,70,91) | 7 |
| | GN-HC | (37.2,54.4,73.9) | 4 | (38,52,70) | 3 | (37,55,74) | 3 |
| | **HGTS** | (——,51.0,——) | 1 | | | | |
| | HMVO | (36.8,54.3,74.7) | 5 | (37,53,74) | 4 | (39,56,72) | 4 |
| | HQPSO | (34.4,52.3,72.2) | 2 | (34,51,72) | 1 | (35,54,73) | 1 |
| | IABC | (37.8,55.8,77.7) | 6 | (36,54,74) | 5 | (42,59,84) | 6 |
| | SNSA | (46.7,68.2,91.0) | 9 | (40,65,93) | 8 | (47,72,93) | 8 |
| | TLBO | (36.1,57.5,78.2) | 7 | (36,55,73) | 6 | (37,61,82) | 5 |
| Case 6 | BS-HH | | | | | | |
| (15 × 10) | CGA | | | | | | |
| | GN-HC | (37.7,55.1,75.1) | 2 | (35,55,71) | 1 | (39,58,77) | 1 |
| | **HGTS** | (——,50.2,——) | 1 | | | | |
| | HMVO | | | | | | |
| | HQPSO | | | | | | |
| | IABC | | | | | | |
| | SNSA | (44.8,65.0,87.8) | 3 | (46,63,83) | 2 | (48,68,89) | 2 |
| | TLBO | | | | | | |

Table 1 shows that GN-HC obtains a ranking varying from second to fifth place in the best average fuzzy makespan among the nine algorithms employed for comparison. GN-HC obtains fourth place overall, only surpassed by BS-HH, HQPSO and HGTS, with a

similar performance to HMVO. All of them are algorithms recognized for their excellent performance. About the best and worst fuzzy makespan, GN-HC has a ranking from second to fourth place in the first five cases and it overtakes the SNSA in the last case. These results show the competitiveness of the GN-HC in solving FFJSSP instances. Figure 5 presents the best fuzzy makespan obtained for each case.

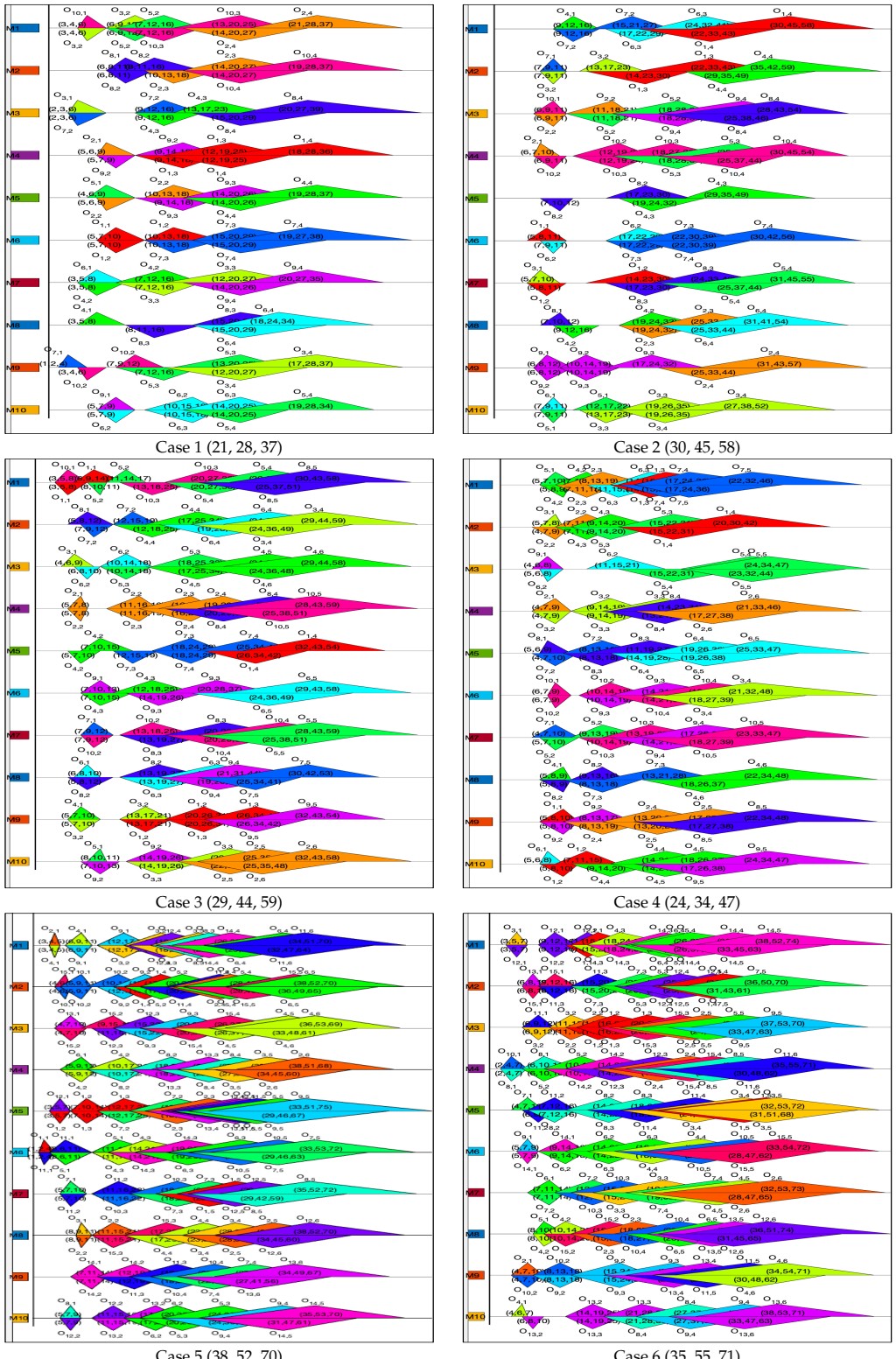

**Figure 5.** The best solution of Cases 1–6 obtained by the GN-HC.

## 6. Conclusions and Further Work

This paper has presented a new algorithm called GN-HC for optimizing the FFJSSP. This algorithm divides the problem into two stages. In the first one, a global neighborhood to search the solution space is created mainly focused on finding the best possible order of operations. In the local search, the machine allocation is tuned through a random-restart hill-climbing to obtain a better smart cell.

The originality of the GN-HC lies in the fact that the global neighborhood has a straightforward implementation and is based on the concurrent application of easy-to-implement operators. On the other hand, the local search is based on hill climbing, improving its performance by focusing on the critical operations that define the fuzzy makespan and using the maximum crisp makespan estimation to discard solutions in a shorter time.

These operations define a simple hybrid algorithm that obtains competitive results compared to current methods recognized for their excellent performance on FFJSSP instances.

The GN-HC can be helpful in real-world applications where the processing time cannot be obtained in advance since many situations may arise during the manufacturing process, such as maintenance, breakdowns, or rush jobs. Future work would be to apply the GN-HC in a real application.

The GN-HC is limited to solving instances of the FFJSSP under the assumption of working in a single factory. It does not consider the case where each instance can be processed in several factories with similar technologies, nor does it consider travel times or inventory capacity, which can be addressed in a later study.

Another proposed study involves utilizing other operations to develop the global search, such as genetic operators. Moreover, other local search strategies can be investigated, such as different implementations of tabu search or simulated annealing. On the other hand, the effectiveness of the proposed method for other types of JSSP variants caused by different types of disturbances can be investigated, such as the arrival of new jobs, revocation of jobs or random transfer times.

In the proposed theoretical research, other kinds of uncertainty criteria can be investigated to select the maximum of two fuzzy values [37], like the one used in the HGTS [4], taking into account the three components of a fuzzy number to select the maximum, in order to manage uncertainty with better consistency.

**Author Contributions:** Conceptualization, N.J.E.-S. and J.C.S.-T.-M.; methodology, N.J.E.-S., J.C.S.-T.-M. and L.J.M.-A.; software, N.J.E.-S. and J.C.S.-T.-M.; validation, L.J.M.-A., I.B.-V. and J.M.-M.; formal analysis, N.J.E.-S., J.C.S.-T.-M. and J.M.-M.; investigation, N.J.E.-S., J.C.S.-T.-M., L.J.M.-A. and I.B.-V.; resources, J.C.S.-T.-M.; data curation, J.C.S.-T.-M. and J.M.-M.; writing—original draft preparation, J.C.S.-T.-M., L.J.M.-A. and I.B.-V.; writing—review and editing, J.C.S.-T.-M., L.J.M.-A. and I.B.-V.; visualization, N.J.E.-S., J.C.S.-T.-M. and J.M.-M.; supervision, J.C.S.-T.-M., I.B.-V. and J.M.-M.; project administration, J.C.S.-T.-M.; funding acquisition, J.C.S.-T.-M. and J.M.-M. All authors have read and agreed to the published version of the manuscript.

**Funding:** This study was supported by the Autonomous University of Hidalgo (UAEH) and the National Council for Science and Technology (CONACYT) with project numbers F003/320109 and CB-2017-2018-A1-S-43008. Nayeli Jazmin Escamilla Serna was supported by CONACYT grant number 1013175. Leonardo Javier Montiel Arrieta was supported by CONACYT grant number 713103.

**Institutional Review Board Statement:** Not applicable.

**Informed Consent Statement:** Not applicable.

**Data Availability Statement:** The GN-HC source code is available on Github https://github.com/juanseck/GN-HC (accessed on 20 September 2022).

**Conflicts of Interest:** The authors declare no conflict of interest. The funders had no role in the design of the study; in the collection, analyses, or interpretation of data; in the writing of the manuscript; or in the decision to publish the results.

## Abbreviations

The following abbreviations are used in this manuscript:

| | |
|---|---|
| JSSP | Job shop scheduling problem |
| FJSSP | Flexible job shop scheduling problem |
| FFJSSP | Fuzzy Flexible job shop scheduling problem |
| GN | Global neighborhood |
| HC | Hill-climbing |
| OS | Operation sequence |
| MS | Machine sequence |

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
