# Peer review of "A Global Neighborhood with Hill-Climbing Algorithm for Fuzzy Flexible Job Shop Scheduling Problem"

_mathematics, doi:10.3390/math10224233_

Round 1

Reviewer 1 Report

Please read the attached file.

Thank you.

Reviewer 2 Report

In this study, the authors propose using machine learning - GN-HC algorithm to solve the FFJSSP. This method explores solution neighborhoods using concurrent insertion, exchange, and path-relinking. For local search, essential operations are first recognized, and one is randomly picked and shifted to a separate processing machine. Backtracking from the final makespan-defining process to the first operation selects essential operations. To speed up hill-climbing, the new crisp makespan uses the longest fuzzy processing time to choose the optimum machine for each function. This paper defines an algorithm for global solution exploration using simple and well-known operations. Hill-climbing uses basic operators to construct a robust, easy-to-implement technique, sped up by selecting essential functions and estimating makespan. The GN-HC compares six previously published methods using the same benchmark with five examples of increasing processes and machines, demonstrating competitive and good results.

1.     Line 176: what Equation did you want to mention or explain?

2.     Figure 6: Please delete the axis X=29, Y=14.5 at the center of the figure.

3.     Run 2 cases more (10 × 15) to compare the other results.

4.     How can you validate the results of this study?

5.     What are the author's recommendations for further study? And what is the limitation of your proposed algorithm? If yes, please describe and suggest future solutions.

Reviewer 3 Report

This paper tackles an extended version of the well-known flexible job shop scheduling problem in which the processing times of the tasks are uncertain. This uncertainty is modelled by means of triangular fuzzy numbers (TFN). In this work the objective considered is the fuzzy makespan. As solution method, the authors propose a hybrid algorithm with a global neighborhood to explore the search space and a hill climbing for performing exploitation. This algorithm is denoted GN-HC.

The topic is interesting, nevertheless, the contribution is marginal, and I’m seriously concerned about several considerations that are in the basis of the paper. Besides, the experimental study is not compelling at all.

More detailed comments are in the following:

1.     Several statements should be justified, some of them in the introduction, 

2.     There are an absolutely lack of references in all the manuscript except in the section of Literature review.

3.     A model of uncertainty, to deserve that name, must model all the possible scenarios. In this kind of problem, a fuzzy schedule must model all possible schedules that can appear with any possible value of processing time compatible with the fuzzy processing time and therefore any actual completion time must be compatible with the fuzzy completion time. This simple and natural fact is not true when the maximum is approximated as it is explained in pg.4 row 140. There are alternative approximations holding this property. I’m aware that several papers use this arithmetic, but this is not reason enough to validate it.

4.     With regards to the test instances, in addition to the ones considered, the authors must experiment with others that are openly available and that were used in other methods in the state of the art; in particular, those considered in [9].

5.     The literature review should serve to identify the state-of-the-art algorithms, but this seems not to be the case here. The authors have not considered the state-of-the art algorithms in their experimental study, and so it cannot be considered compelling at all.  They considered CGA [32], but and they do not considered HGTS [9], which showed better results than CGA and, by the way, produces better results than the proposed GN-HC as well in most of the cases.

Some minor issues:

Pg. 3, r 97: citePalacios2015b….

Pg.5, r 176: Eq. ??

Pg.5 r. 181 and 191 the mutation for machine assignment is repeated (not using the same words, but it is mostly the same)

Round 2

Reviewer 1 Report

Dear Authors and Editors;

The authors have corrected and answered all my comments and questions carefully. However, please revise the format of reference [21]. The reviewer strongly suggests that the manuscript should be accepted for publication.

Thank you. 

Sincerely yours, 

The reviewer.

Reviewer 3 Report

The authors have considered several of my suggestions;  however, in the most important ones, their answers are not convincing.

As the authors say “Equation 2 is a criterion to compare two fuzzy numbers and to decide which of them is the largest one.” I agree, but this is a ranking for them. The objection of my review was not with the ranking, but with the use of this ranking to approximate the maximum, which is where the uncertainty model loses consistency, in my opinion. The fact that “is widely used in the works” is, as I have mentioned in my first review, not a reason to support its use. It is not true that “changing this criterion would prevent a proper comparison against the other methods”; notice that HGTS is built using another approximation to the maximum and it is compared with other methods (although in these comparisons the results of HGTS are at a disadvantage compared to the other methods, since such approximation of the maximum produces greater or equal values ​​in the three components of the TFN).

The last sentence of the conclusions is not a satisfactory solution in my opinion.

The response to the suggestion to extend the experimentation to instances proposed in [9] is also unconvincing. They have not considered these instances because there are only results from HGTS and, in their words “this would require running all algorithms to generate a fair comparison”. However, one more instance of Lei set is considered (Case 6), eventhough only HGTS and SNSA report result for it. In this new instance, GN-HC fell below HGTS and outperformed SNSA.

Taken also into account that on the six instances tested, GN-HC is worse than three of the nine algorithms, one might wonder if the rest of the instances have not been tested because they cannot present competitive results on them.
